# A New Census of Protein Tandem Repeats and Their Relationship with Intrinsic Disorder

**DOI:** 10.3390/genes11040407

**Published:** 2020-04-09

**Authors:** Matteo Delucchi, Elke Schaper, Oxana Sachenkova, Arne Elofsson, Maria Anisimova

**Affiliations:** 1ZHAW Life Sciences und Facility Management, Applied Computational Genomics, 8820 Wädenswil, Switzerland; delt@zhaw.ch (M.D.);; 2Swiss Institute of Bioinformatics, 1015 Lausanne, Switzerland; 3Science of Life Laboratory, Department of Biochemistry and Biophysics, Stockholm University, 106 91 Stockholm, Sweden

**Keywords:** tandem repeat, homorepeat, domain repeat, protein repeat, repeat prediction, intrinsic disorder, protein function, Swiss-Prot

## Abstract

Protein tandem repeats (TRs) are often associated with immunity-related functions and diseases. Since that last census of protein TRs in 1999, the number of curated proteins increased more than seven-fold and new TR prediction methods were published. TRs appear to be enriched with intrinsic disorder and vice versa. The significance and the biological reasons for this association are unknown. Here, we characterize protein TRs across all kingdoms of life and their overlap with intrinsic disorder in unprecedented detail. Using state-of-the-art prediction methods, we estimate that 50.9% of proteins contain at least one TR, often located at the sequence flanks. Positive linear correlation between the proportion of TRs and the protein length was observed universally, with Eukaryotes in general having more TRs, but when the difference in length is taken into account the difference is quite small. TRs were enriched with disorder-promoting amino acids and were inside intrinsically disordered regions. Many such TRs were homorepeats. Our results support that TRs mostly originate by duplication and are involved in essential functions such as transcription processes, structural organization, electron transport and iron-binding. In viruses, TRs are found in proteins essential for virulence.

## 1. Introduction

The continued progress in molecular biology demands better classification and understanding of genomic sequences, their evolution and function across the tree of life. Proteins indisputably remain at the heart of the molecular machinery performing a multitude of essential functions. According to most recent estimates a substantial amount of proteins contains adjacently repeated amino acid (AA) sequence patterns, known as tandem repeats (TRs). TRs are described by the length of their repeating motif (unit length *L*), the number of repeated units (size *n*) and the similarity among their units [1] (Figure 1).

TRs are highly abundant in the human proteome (e.g., according to [2] >60% proteins contain TRs) and display an impressive variability of sizes, structures and functions [2,3,4]. Proteins containing TRs have enhanced binding properties [5] and are known to have associations with immunity-related functions [6,7] and diseases. For example, amyotrophic lateral sclerosis, myotonic dystrophy, dentatorubral-pallidoluysian atrophy, frontotemporal dementia, fragile X syndrome, Huntington disease, spinobulbar muscular atrophy and spinocerebellar ataxia are all caused through tandem repeat disorders (TRD) [8].

Newly appeared TRs are typically perfect unit copies, but evolution erodes the similarity between the TR units through indels and point mutations, duplication and loss of TR units, recombination, replication slippage and gene conversion [9]. TR regions provide a rich source for genetic variability with a wide range of possible genotypes at a given locus [10]. TRs are prone to selection on long evolutionary scales as well as on a somatic level. The occurrence of mutations in TR within protein coding genes, can alter the structure and therefore likely the function of the affected protein too. Since non-coding regions play crucial roles in gene regulation, transcription, and translation, the proteins concerned are also likely to be affected by selection. While the biological mechanisms generating TRs are not well understood, evidence suggests that natural selection indeed contributes to shaping TR evolution [3,11], and that TR expansion is linked to the origin of novel genes [12]. TRs have been successfully exploited in bioengineering due to their “design-ability” [13].

### Comprehensive Annotation of Proteomic Tandem Repeats

Despite much interest [8,14,15,16], the most recent and commonly cited census of protein TRs summarizing repeats in the curated protein knowledge base UniProtKB/Swiss-Prot [17] dates back two decades [18]. Since then this popular data bank has grown more than seven-fold (Appendix A). Equally, a multitude of new methods were developed for the prediction and analysis of TRs. In fact, TRD’s definition often lacks a biological interpretation and is based rather on empirical properties of TR unit similarity [19,20,21]. In particular, due to striking differences in TR predictor properties and definitions, a new statistical framework and a meta-prediction approach was proposed in order to increase the accuracy and power of the TR annotation which filters annotations based on a biologically meaningful mathematical model [2,19].

Here, we apply this recent methodology to characterize the distribution of protein TRs in the UniProtKB/Swiss-Prot protein knowledge base [17,22]. Our TR annotation for each protein includes the TR region start, end, minimal repeated unit length, among unit divergence and TR unit alignments. This allows our study to provide an unprecedented detail of the universe of protein TRs. We respond to the call [14] and apply the state-of-the-art method for TR detection followed by filtering through a sound statistical framework.

To our knowledge, no recent study connected the information of viral proteomic TRs to their hosts TRs. Viral genomes are generally relatively small which demands for an optimal coding capacity. Furthermore, they lack their own translational apparatus and depend completely on their hosts protein synthesis machinery [23,24]. We compare the TR distributions in virus and host proteomes to identify putative similarities which could bring insights into viral proliferation.

Proteins with TRs tend to be enriched with intrinsic protein disorder (IDP) [25], and vice versa [26]. Intrinsic disorder has been found in different regions of proteins with different structures, enabling an array of functions like recognition domains, folding inhibitors, flexible linkers, etc. [27,28]. Both TR and intrinsic disordered regions (IDR) tend to be over-represented in the hubs of protein-protein interaction networks [29,30]. However, while the relationship between these non-globular protein features was observed, the biological reasons are not well understood. TRs often fold into specific structures, such as solenoids, or have “beads on a string” organizations [4]. However, there is undoubtedly a class of protein TRs strongly associated with unstructured regions [25,26]. Several studies have shown that compositionally biased, low complexity regions often found in IDPs, evolve rapidly, including recombinatorial repeat expansion events [31,32]. Others in contrast observed that the association between repeat enrichment and protein disorder is not as clear [33]. To systematically characterize and explore the enigmatic connection between TRs with IDP, we annotate each protein with IDP regions and summarize the distribution of the overlap of TR and IDP regions over all kingdoms of life.

## 2. Materials and Methods

### 2.1. Tandem Repeat Annotations

Tandem repeats were annotated using TRAL [1], which validates the results from several popular TR predictors, clusters redundant and overlapping regions, removes false positives and annotates known TRs [19]. Assuming that TR units should come from a common ancestral unit [2], TRAL compares the null hypothesis that the proposed TR units are evolutionary unrelated (i.e., infinite time to the common ancestor) against the alternative hypothesis that they are evolutionary related by duplication (i.e., finite time to the common ancestor). This is done using the likelihood ratio test, which was shown to control the rate of false positive TR annotations as defined by the model-based statistical significance level, here (α=0.01). By default, TR predictors used by TRAL were T-REKS [34], XSTREAM [35] and HHrepID [36]. T-REKS and XSTREAM perform best for the detection of short TRs, while HHrepID is best at detecting domain TRs. PFAM domains occurring in tandem were annotated using the corresponding sequence profile models, which were retrieved from PFAM [37] and converted to circular profile models [3].

All de novo and PFAM annotations of TRs were subjected to a filtering step (p=0.1, $n_{effective}>1.9$, e.g., see [19]). Next, for every sequence, the overlap of TR annotations was determined. To retain small TRs within domain TRs, or TRs overlapping only in their flanks, the overlap was not determined by shared amino acids. Instead, a shared ancestry criterion was used: If two TR predictions shared any two amino acids in the same column of their TR multiple sequence alignment, they were considered to be the same TR. In this case, the de novo TR (in a tie with a PFAM TR) or the TR with lower p and higher divergence (in a tie between two de novo TRs) was removed.

To homogenize and refine all remaining de novo annotated TRs, they were converted to circular profile hidden Markov models, re-annotated [3], and once again filtered using a more stringent *p*-value threshold (p=0.01).

The pipeline was implemented in Python3 and all code is available on github.com/acg-team/swissrepeats.

### 2.2. Annotation of Intrinsic Disorder

Intrinsic disorder often causes difficulties for experimental studies of protein structure, making proteins very difficult to crystallize. Even if X-ray crystals can be obtained or structure described via nuclear-magnetic resonance imaging (NMR), these data may be challenging to interpret. Taking into account known specifics of IDRs, various computational methods have been developed to label each amino acid in a protein sequence as ordered or disordered. These methods, however, are limited to recognize patterns observed in experimentally annotated disorder and each predictor is tailored to identify a certain type of characteristics. There is no standard definition of disorder and no large set of universally agreed disordered proteins. Moreover, different parts of proteins can be ordered or disordered under different conditions. It is, therefore, important to carefully annotate using different definitions of disorder.

Disorder annotations have been extracted from MobiDB [38] covering 546,000 entries of UniProtKB/Swiss-Prot (Release 2014_07). MobiDB provides consensus annotations as well as raw data from DisProt, PDB (missing residues in X-Ray and NMR) and 10 computational predictors: three variations of ESpritz [39], two variations of IUPred [40], two variations of DisEMBL [41], GlobPlot [42], VSL2b [43,44] and JRONN [45,46]. We annotated disordered residues using per-residue probability scores with a cut-off of 0.5.

### 2.3. Analysis of Tandem Repeats and Intrinsic Disordered Regions Annotations

The data analysis pipeline was implemented in R version 3.6.0 (2019-04-26) and is available on github.com/acg-team/swissrepeats. Therefore, the data is easily reproducible and can be used for follow-up studies.

#### 2.3.1. Tandem Repeat Location Normalization

For TRs that cover a significant fraction of a protein, their center necessarily falls near the middle. To avoid such bias of a simple center/length metric, TR location was normalized by allowing only the valid center locations:(1)$$x=\frac{C-\frac{l\cdot n}{2}}{N-l\cdot n}$$
where *N* is the total length of the protein sequence, and the length of a TR region is the product of the effective number of TR units *n* and the TR unit length *l* (for definitions see [3]). *C* is the middle of the TR of size l·n. We required the denominator to be >0.

#### 2.3.2. Expected Number of Homorepeats

To compare expected and empirical number of homorepeats in Swiss-Prot, we derived the expected number of homorepeats of an amino acid *a* with *n* repeat units in a random sequence of length *s*, given the AA frequency p(a). Mathematically, this problem corresponds to the probability of sequential runs of successes in a Bernoulli trial, where p(a) is the probability of a success, and so the expected values and variances can be derived for all sequences or subsequences of different lengths in the sequence set. Exact solutions to the expected value and variance of the number of runs of a given length in a bounded sequence of length are derived in, e.g., [47].

#### 2.3.3. Correlation Analyses

To assess the relationship between the mean protein length and the fraction of proteins with TRs, we performed a one-sided Pearson product-momentum correlation coefficient test for each type of TRs separately. The null hypothesis that there is no correlation $H_{0}:\rho_{0}=0$ was tested against the alternative hypothesis $H_{A}:\rho_{A}>0$ for observing a positive correlation at the significance level α=0.001.

To assess the relationship between the AA disorder propensity [48] and the AA frequency (separately, in TRs, in all proteins and in all proteins without TRs), we performed a two-sided Spearman’s rank correlation analysis at the significance level α=0.05.

To assess the correlation analysis of TR content in Viruses compared to their host organisms, we performed a five-sample test for equality of proportions [49] without continuity correction and at the significance level α=0.05. The null hypothesis tested was that the proportions of unique TRs per protein in the two groups (virus-host vs. viruses) are the same for each superkingdom and independently in human.

## 3. Results

Exhaustive annotation of protein TRs in the entire UniProtKB/Swiss-Prot was done using a meta-prediction approach based on both de novo and hidden Markov models (HMM) profile-based methods followed by filtering of false positives and redundancies. The pipeline was implemented in Python using TRAL [1]; see Methods for details. Structural and biochemical properties of TRs can be extremely diverse depending on the length and the composition of their minimal repeating unit. We studied TR properties in four categories defined according to TR unit length *L*: (1) homorepeats L=1, (2) microrepeats 1<L≤3, (3) smallrepeats 4<L≤15, and (4) domainrepeats ≥15. Please note that small variations in exact category boundaries did not have much influence on our conclusions. We chose a minimum of 15 residues for domain TRs based on [3] who showed it as a necessary minimum length to detect evolutionary signal.

Table 1 summarizes the predicted TR annotations and their distributions with respect to protein length and repeat number by superkingdoms [50].

### 3.1. Tandem Repeats Are Abundant in Proteins of All Domains of Life

Overall, Bacteria had the most entries in Swiss-Prot but Eukaryota had the biggest fraction of proteins with TRs: 50.9% of all eukaryotic proteins contained at least one TR-68.8% in *Homo sapiens*, 61.9% in *Mus musculus*, and 60.8% in *Drosophila melanogaster*. Interestingly, 43.6% of viral proteins contained TRs, almost as frequently as in Eukaryotes. In comparison, fewer prokaryotic proteins contained TRs, but nevertheless TRs were found in >30% of both bacterial and archaeal proteins and, for example, in 28% of proteins in *Escherichia coli*. This is in alignment with the findings from Lavorgna et al. [16] detected on the much smaller data set and without the same stringent statistical evaluation.

Homo TRs made up 20% of all annotated TRs over all superkingdoms and 30% for human TRs. Of all homo TRs 91.3% were of eukaryotic origin. Homo TRs were usually annotated as short regions (mean = 8.8 units). We could not detect a protein which contained only homo TRs and no other type of repeat.

56% of all annotated TRs were micro TRs, with a mean of 7 units.

76% of all annotated TRs were small TRs, which on average had fewer units than either homo or micro TRs (mean = 6 units). Proteins containing solely small or micro TRs, had on average only three TR units.

30% of all predicted TRs were domain TRs with a mean of 3.5 TR units. Among prominent exceptions is an extracellular matrix-binding protein (UniProtKB ID = Q5HFY8, *Staphylococcus aureus*) with 80 units each of 97 AA (UniProtKB entry ID = PF07564) and spanning overall a TR region of ≥7700 AA. Other examples of bacterial domain TRs with many units are the cell surface glycoprotein 1 of *Clostridium thermocellum* and uncharacterized PE-PGRS family proteins of *Mycobacterium tuberculosis* with almost 300 TR units, 6 residues each.

Eukaryotic domain TRs tend to be more uniformly distributed in terms of their TR size. The proteins with the most repeat units belong to mediator of RNA-polymerase II transcription subunit proteins from yeast (*Eremothecium gossypii*), slime mold (*Dictyostelium discoideum*) and human Mucin-22 protein. Among viral TRs, most TR units were observed in collagen-like proteins of the Mimiviridae.

### 3.2. Vast Variation in Tandem Repeats Lengths and Unit Copy Numbers

Tandem repeats are not homogeneously distributed in terms of their unit lengths and numbers. Figure 1 reveals multiple peaks, showing that some unit lengths are particularly frequent. These peaks represent common TRs, with specific TR units used in different proteins in varying number. Among prominent examples are zinc-finger TRs, abundantly present in proteins of all domains of life, but also LRRs and WD40-like beta propeller repeats.

Eukaryotes tend to have the longest TR units, among extreme outliers are the Anchorage 1 protein and Nesprin homolog in *Caenorhabditis elegans* and Mucin-12 and FCγBP (which has mucin-like structure [51]) in humans.

In Bacteria extreme outliers in terms of unit length were detected for example, in the Alzheimer disease causing bacterium *Porphyromonas gingivalis* [52] and in the Mannuronan epimerase protein of *Azotobacter vinelandii*. In hemagglutinin A of *P. gingivalis*, which plays role in host colonization by adhesion to extracellular matrix proteins and which is expected to be involved in periodontal diseases [53,54], a TR with L>450 was annotated.

### 3.3. Multiple Tandem Repeats per Protein

A substantial fraction of proteins contained more than one distinct TR region, most frequently in eukaryotic proteins (56% of all proteins with TRs), but also in viral (45.7%) and prokaryotic proteins (28.4% in Bacteria and 26.6% in Archaea). In Eukaryotes, 43% (90026 absolute count) of all proteins with TRs had ≥4 distinct TR regions. Viruses with 28.6% of proteins with TRs having ≥4 distinct TR regions were followed by Bacteria with 9.1% and Archaea with 8.0%.

Most proteins annotated with multiple individual TRs belong to structural (i.e., Titin, Fibroin, Mucin, Desmoyokin) and LRR-containing eukaryotic proteins, to pathogenesis associated bacterial proteins responsible for bacterial binding i.e., to platelets (Serine-rich adhesin for platelets) and to fibronectin (Extracellular matrix-binding protein ebh), to nuclear antigen, transcription factor and structural viral proteins and to archaeal proteins involved in mucin and DNA break repair. It seems possible that some continuous TR regions may have been detected as separate TRs due to the lack of power of TR detection in certain regions, such as for divergent TRs. However, even if this explanation holds true for certain cases, it is not generally applicable. In proteins annotated with ≥4 TRs, the TR types were not necessarily the same. By far the most frequent TRs in proteins containing ≥4 TR regions, were smallrepeats (95.0% of all predicted TRs), followed by microrepeats (87.9%), and domainrepeats (47.6%) (Appendix A).

### 3.4. More Tandem Repeats Are Found in Longer Proteins

We observed that chloroplastic proteins tend to be shorter than mitochondrial proteins, and had less annotated TRs. Non-mitochondrial and non-chloroplastic proteins were on average longer and contained more TRs. Figure 2a displays a linear relationship between mean protein length and the number of TRs. Prokaryotic proteins cluster with their protein length and TR content in the same range as chloroplastic proteins. It seems that TRs are increasingly abundant in increasingly complex organisms (consider also Appendix A).

A different level of detail was obtained by grouping the number of TRs for different ranges of protein sequence length. Figure 2b shows that in general, differences in TR distributions observed between kingdoms can be largely attributed to protein sequence length with Eukaryotes having on average more TRs, longer proteins and the longest TRs over all superkingdoms. Eukaryotic proteins with homo TRs displayed a different distribution compared to other TR types and to the other superkingdoms. The amount of homo TRs in eukaryotic proteins increases exponentially whereas for the other superkingdoms, the homo TR fraction remains similar to increasing protein length (Appendix A). In proteins with TRs, longer protein sequences tend to have on average more TRs and small TRs seem to be the most recurrent in all kingdoms (Appendix A). It might be that small TRs benefit from a good trade-off between size and energy investment required for duplication. Overall, we observe a positive correlation between the protein length and the fraction of proteins with TRs across all kingdoms of life for all TR types: $R^{2}=0.46,\;p<0.001$ for homo TRs; $R^{2}=0.64,\;p<0.001$ for micro TRs; $R^{2}=0.88,\;p<0.001$ for small TRs, and $R^{2}=0.24,\;p=0.08$ for domain TRs.

The lack of correlation for domain TRs, where factors other than protein length must contribute, could be explained by differences in TR generating processes for different TR types, with domains acting more like architectural blocks performing certain functions on their own. On the other hand, consistent with the same trend, we observed that homorepeats are particularly frequent in Eukaryotes, where proteins were on average longer. Moreover, longer homorepeats are mostly characteristic to eukaryotic proteins, e.g., see Figure 3. PolyQ and polyN homorepeats may often be observed with >50 repetitions in Eukaryotes. The same homorepeats display <10 repetitions for polyQ and <20 for polyN in Prokaryotes and Viruses. This large discrepancy cannot be explained purely by the length of the involved proteins.

### 3.5. Tandem Repeats-Location Is Biased towards the Flanks for Shorter Tandem Repeats

Next, we explored where in a protein sequence TR regions tend to be found. The location within a protein was evaluated with respect to the center of a TR region and normalized by the protein length (see Methods). The observed distribution of TRs along the protein length was non-uniform and dependent on the TR unit length. Appendix A shows the distributions of the relative positions of TRs in proteins across all different kingdoms and for different TR unit length categories.

As expected, TRs appear more frequently near the start of protein sequences. For homorepeats such tendency was particularly striking, especially in Archaea, where most homorepeats were found in the C-terminus while most domain TRs were at the N-terminus (Appendix A). Overall, shorter TRs (homo, micro and small) displayed stronger preferences towards both N- and C-terminals of proteins. In particular for Eukaryotes, there was a clear correlation between the TR unit length and the location bias towards the protein flanks. Interestingly, also domain TRs in Viruses and to a smaller degree in Archaea tend to be located towards both flanks of proteins. In contrast to Archaea and Bacteria, in eukaryotic proteins TRs were found to be over-represented in the N-terminal protein flank.

Analogously we examined the location of IDRs in proteins and found similarities to the location of TRs. IDRs, also, were found more frequently towards the flanks, while short IDRs were often located near the N-terminal (Appendix A).

### 3.6. Tandem Repeats Are Enriched with Disorder-Promoting Amino Acids

Furthermore, we evaluated the abundance of amino acids in TRs with respect to their disorder-promoting propensity [48].

The amino acid abundances in TRs were linearly dependent to their respective abundances in proteins (Figure 4). TRs could not be characterized by a certain amino acid abundance pattern. However, we observed a significant positive correlation (ρ=0.71,p<0.05) of AA abundance in TRs with the corresponding AA disorder propensity (Appendix A). In contrast, the overall AA abundance in all proteins (incl. TR-containing) showed much lower correlation (ρ=0.44,p=0.053). We could not detect any correlation when the TR-containing proteins were excluded from the overall fraction (ρ=-0.10,p>0.05).

Homo TRs were on average longer in Eukaryotes compared to the other superkingdoms (Figure 3). In homo TRs annotated as IDRs, AAs with a higher disorder-promoting propensity were more abundant since these homorepeats were typically longer. This indicates that long homorepeats of disorder-promoting AAs tend to be disordered or at least tend to be predicted as such.

In general, we observe an exponential decay of homo TR number with increasing number of repetitions with some exceptions. PolyN and polyQ showed a different decay pattern compared to other homo TRs – they were much more frequently observed as longer stretches. PolyN and polyH, repeats of AAs with relatively low disorder-promoting propensity, appeared frequently as long repeats. PolyN and polyH repeats were frequently detected within regions annotated as disordered, even though their disorder-promoting propensity is relatively low Figure 5a).

We compared empirically observed numbers of homo TRs with their statistical expectation (estimated as sequential runs of success in a Bernoulli trial; see Methods) (Figure 5b). Disorder-promoting amino acids appeared more frequently than expected in longer homorepeats.

### 3.7. Tandem Repeats and Intrinsic Disordered Regions Are Often Found in the Same Protein

To elaborate the association of TRs with intrinsic disorder, we distinguished four types of overlap (Appendix A). We called the overlap a “tail-overlap” where an IDR began within the TR region and ended after the end of the TR region. In contrast, we called it a “head-overlap” if an IDR began before a TR region and ended within. A scenario where the IDR was completely within a TR region was called *Disorder-in-TR*, whereas a scenario where a TR region was completely within an IDR was called *TR-in-Disorder*. Among all residues in UniprotKB/Swiss-Prot, 9.1% were in TR regions, 12.9% were in IDRs and 2.5% were in the overlap (Figure 6B). Overall, few residues were annotated to be in both TR and IDR. However, both TRs and IDRs were frequently found in the same protein. IDRs were found in 27.8% of proteins with TRs, whereas 19.6% of the IDRs overlapped with TR regions. Most overlaps were due to cases where a complete TR region was within an IDR (Figure 6A).

### 3.8. Tandem Repeats Are Involved in Transcription Processes, Structural Organization, Electron Transport and Ion-Binding

From all detected TRs, 4.5% were annotated using a PFAM profile model. Many more PFAM-based TRs were detected in Eukaryotes (73%) compared to the other superkingdoms.

For each superkingdom, here we list ten most frequently observed PFAM families occurring as TRs in proteins (the whole list is in Appendix A). Overall, we detected many TRs in proteins which are involved in transcription. As expected, in Eukaryotes many Zn-finger TRs were responsible for adhesion to DNA, RNA and lipids. WD-40 repeats are inter alia involved in transcriptional regulation. RNA recognition motifs, such as the K homology (KH) domain which binds to RNA, is involved in transcriptional repression and like the Pumilio family, which occurs in many TR-containing proteins in fungi. Eukaryotic proteins with TRs are frequently involved in RNA-polymerase binding and RNA-splicing by e.g., KRAB box domain, as well as in the assembly of multiprotein complex.

Additionally, many TRs were detected in proteins involved in electron transport and ion-binding such as EF-hand domain pair and Ca^2+^-binding EGF domain in Eukaryotes, the zinc-dependent enzyme UDP N-acetylglucosamine O-acyltransferase and the Rad50 zinc hook motif in Bacteria. Zinc-binding motifs could be detected in viral TR-containing proteins such as the zinc knuckle motif, too.

In proteins of chloroplastic origin, we detected many TRs in NifU-like domains. They are involved in the formation of metalloclusters of nitrogenase in certain bacteria and the maturation of FeS clusters.

Many TRs were detected in proteins of mitochondrial origin with EF-hand domain pairs, found in a large family of Ca^2+^-binding proteins, and with bacterial transferase hexapeptide which combines several transferase protein families including zinc metalloenzymes. Both in chloroplasts and mitochondria, many pentatricopeptide repeats (PPR) were detected. They play roles in RNA stabilization and processing. We further found ankyrin repeats (especially in viruses), which are known for their diverse functions in transcription- and cell-cycle regulation, signaling and ion-transporters, but also have cytoskeletal functions.

Proteins involved in the structural organization of cells can be found in all superkingdoms. Well known are the extracellular structure proteins of the collagen superfamily involved in formation of connective tissues, but also leucine-rich repeats (LRRs) which are unusually rich in hydrophobic AAs forming a solenoid protein domain. They seem to provide a structural framework for the formation of protein-protein interactions [55,56]. Proteins with LRRs are involved in transcription, RNA processing, signal transduction and more [57].

In Eukaryotes we further found many TRs in proteins with LIM domains and tetratricopeptide repeats (TPRs). LIM domains are formed by two zinc-finger domains and are involved in cytoskeletal organization, organ development and oncogenesis. Both TPR and LIM motifs mediate protein-protein interactions. TPRs also play roles in cell-cycle regulation, transcriptional control, protein transport, neurogenesis and protein folding.

We further observed that many TR-containing proteins of chloroplastic origin contained the catalytic domain of homoserine dehydrogenase involved in the aspartate pathway, which leads to the production of AAs as well as essential components of bacterial cell wall biosynthesis [58,59].

For TRs in proteins which could be associated with a protein family, we calculated the TR center location (see Methods) of the most frequent PFAMs in each superkingdom shown in Appendix A. TRs in proteins containing the ribosomal protein L6 domain appeared to be located at the same position in Archaea and Eukarya. A similar trend was observed for transferase hexapeptide in Archaea and Bacteria, for WD-40 beta propeller repeat in Bacteria and Eukaryota, as well as for the TFIIB zinc-binding domain in Archaea, the C2H2-type zinc finger in Eukaryota and zinc knuckle in Viruses. Examining closer the inter-kingdom relationship of the zinc-binding domain, it is interesting to note that in Archaea the N-terminal zinc ribbon is part of the recruitment of RNA-polymerase II, where a beta sheet structure of cysteine and histidine residues coordinates the zinc ion. Similarly, in the viral zinc knuckle domain, a beta sheet of cysteine and histidine mediates the zinc ion. The zinc-finger domain in Eukaryotes is the best described one. Multiple zinc-finger domains appear as tandem repeats building together the DNA-binding domain of the protein by binding into the major groove of the nucleic acid double helix structure.

This does not only show that TRs of proteins with similar function seem to cluster at the same position in the protein across all superkingdoms but also supports the hypothesis of TRs are directly involved in binding activities to nucleic acids and are therefore involved in transcriptional regulation. It is known that long functional IDRs are the binding regions of nuclear proteins, which cluster mostly in hubs [30]. Together with the observation that TRs frequently appear in hub proteins and the fact that they correlate with IDP, this is supportive for our hypothesis of the direct involvement of TRs in small molecule interactions (such as nucleic acid binding or PPI).

### 3.9. Viruses Have Fewer Tandem Repeats Compared to Their Hosts

The TR types were distributed in similar proportion for Prokaryotes (2:4:1 for micro:short:domain) and between Eukaryota and Viruses (1:3:∼1 for micro:short:domain). Therefore, we searched for further TR content similarities specifically between Viruses and their hosts.

Of all Swiss-Prot proteins, only 3% were viral. Majority of the represented viruses had a eukaryotic viral host (92%) followed by bacterial (6%) and archaeal (2%) viral hosts. Of these viral proteins, 43.6% were annotated with at least one TR and 58.7% of these had only a single TR per protein. Only a negligibly small part of them did not have an annotated viral host species.

Most of the viral proteins (95.4%), had only a single associated host species. Some viral proteins were associated with up to 23 different host species, for example, these were capsid proteins and some replication associated proteins. 95.1% of viral TR-containing proteins had a eukaryotic host organism but only a few had a bacterial (3.8%) or archaeal (1%) host.

Figure 7 shows numbers of TRs per protein in viruses and their host organisms. Since human proteins and their “virobiome” [60] are intensively studied, we also compared the TR content in human proteins to human viral proteins (Figure 7D). While there seems to be no clear association, there is a tendency in viruses to have a slightly smaller percentage of sequences with at least one TR when compared to their host. This may point to the dependency between the virus and the host TR creation mechanism. Furthermore, viral protein TRs may be involved in functions very specific to viruses and their ability to infect. For example, viral proteins with TRs are enriched with suppression of host gene expression, transcription, capsid formation (surface markers) and reproduction (attachment to host-cell-receptors, integrase, surface markers).

## 4. Discussion and Conclusions

This study provides an overdue update to the census of protein repeats in Swiss-Prot [18], which has grown more than 7-fold since 1999. Overall, we estimate that just over 50% of proteins contain at least one TR region, substantially more compared to 14% originally detected by Marcotte et al. We attribute this difference mainly to the increased power of TR detection. In contrast to the original study (which relied on one TR prediction method based on self-sequence alignment), we used a meta TR prediction approach based on several state-of-the-art methods, and followed by refinement and statistical filtering steps to control the rate of false positives. Furthermore, our method not only identifies TR regions but also estimates the length of the TR unit and number of repetitions. Thus, our study, brings an unprecedented level of detail about TRs and their interplay with other protein features, such as intrinsic disorder.

Consistent with Marcotte et al., we observed that Eukaryotes and Viruses tend to have more TRs compared to Archaea and Bacteria. Similarly, we could verify the positive correlation between protein sequence length and the number of TRs, also for the larger Swiss-Prot database. Eukaryotic proteins tend to have more than one specific TR per protein. This is consistent with evidence that TR regions are involved in gene regulation and signaling, e.g., [61,62]. Possibly, more complex organisms require more genes to be regulated and more signaling is required, and thus more TRs related to these functions are found in eukaryotic organisms.

Compared to Marcotte et al., we were able to detect many more TRs with shared ancestry between Eukaryotes and Prokaryotes. Indeed, such remote protein homologs are separated by large evolutionary distances, making their detection challenging. We were able to detect these additional cases, since our approach includes a method based on circular profile HMMs. According to our analyses, proteins with chloroplastic or mitochondrial origin cluster together with prokaryotic proteins in regard of their mean protein length and TR content. Eukaryotic proteins with endosymbiotic origin (mitochondria and chloroplast) tend to be shorter and contain fewer TRs. Proteins with origin in chloroplasts are even shorter and with fewer TRs than mitochondrial proteins. PPR repeats are over-represented in eukaryotic proteins with endosymbiotic origin as well as in Prokaryotes. Based on the above, we can suggest that some proteins with TRs of ancient origin are involved in crucial mechanisms in Prokaryotes, while they remain and are presumably functional in endosymbionts. Lavorgna et al. [16] showed support for this hypothesis, applying a different approach. They analyzed specific endosymbiotic relationships and could show a decrease of repeats in organisms with lower complexity than their ancestors (reductive evolution).

Our study shows vast variability of TRs, not only in terms of the characteristics of repeated units but also in terms of numbers of repetitions. The impact of deletions or insertions of TR units or their parts into the already conserved functional proteins must also vary from protein to protein. For example, TRs where individual units fold independently may be more prone to be disordered often displaying functionally important linear motifs [63].

### 4.1. Tandem Repeats Originate through Duplication

In our analysis of the overlap between TRs and IDRs, only 2.5% of all residues were annotated as both—a TR and an IDR. In most overlaps, TRs fall fully within an IDR. TRs have significantly more AAs associated with increased disorder propensity than protein sequences without TRs. In general, both TRs and IDRs lack hydrophobic AAs. IDRs are thought to be unable to form well-organized hydrophobic cores that make up structured domains. However, TRs are known for both—being unstructured but also as folding into specific 3D shapes [4,64]. We therefore hypothesize that if intrinsically disordered sequences are repeatedly duplicated, they may gain the ability to fold in specific tertiary structures contributing to function. In fact, it is known that IDRs are characteristic of eukaryotic protein-protein interaction (PPI) hub proteins [65]. We found TRs and IDR often in the same protein but not often overlapping. This, together with the observation of the occasional overlap of TR and IDR supports the hypothesis of TRs being a typical feature of PPI by i.e., playing a crucial role in the interaction through equipping proteins with a contact side. In addition, conserved disorder regions evolve more rapidly than regions with defined structures and are known to show good properties in binding nucleic acids and in protein-protein interactions [66]. By duplication of such regions, their properties could be modulated resulting in the evolution of new TRs. To support this by future investigations, a comprehensive analysis of PPI-networks should result in a skewed abundance of TRs towards hubs compared to random networks.

The fact that no IDRs were found in mitochondrial proteins [67,68,69] supports our hypothesis that certain, crucial TRs were generated before endosymbiosis of Prokaryotes and persisted within the proteins. If initially disordered regions were accumulated in proteins through alternative splicing or exonization, and by duplication established a stable tertiary structure, they might have been missed by the disorder detection methods.

Furthermore, Marcotte et al. reasoned that repeat expansion requires less energy than the initial repeat formation and that long repeats are preferentially duplicated. Indeed, we observed an increased number of TRs in proteins known for rapid expansion and diversification such as WD-40 domains and Ribosomal L6 protein family [70,71].

Small TRs are most recurrent over all superkingdoms. This might be because the small size seems to be a good trade-off of TR unit length and energy investment in duplication [72].

### 4.2. Tandem Repeats Are Involved in Housekeeping Proteins

Previous studies found that TRs play roles in a diversity of molecular functions, e.g., [3,73]. In our study, TRs appear to be important to the function of many proteins, which are vital to survival. For example, such TRs may be among some striking outliers we observed (in terms of TR unit length and number), which belong to host-colonization proteins of bacteria and viruses. Furthermore, TRs were found to be enriched in proteins with functions in binding DNA and RNA [74]. We could show that the TR location can not only be associated with the location of binding domains but also correlates with the location of enriched intrinsic disorder. IDRs located in the N-terminus are common in DNA-binding proteins [75]. C-terminal IDRs are associated with transcription factor repressor and activator activities. We observed that the TR locations in proteins from families involved in those mechanisms corresponds to the position of IDR. In general, TRs tend to be found towards the N-terminal end of protein sequences. However, micro- and small TRs cluster at both termini. This finding might be explained by the fact that domain TRs tend to be near the N-terminal end, hence they pull the location-distribution of TRs in a protein sequence to the N-terminus. More than half of proteins with TRs, have more than one distinct TR and the TR type can vary within the protein. This might be due to different binding sites and/or different patterns on binding sites of proteins in hubs of protein-protein interaction networks.

### 4.3. Disorder-Promoting Homorepeats Are Longer Than Order-Promoting Homorepeats

Finally, homorepeats deserve special attention as these are widely known for associations to many human diseases. In our study, homorepeats form a large proportion of TRs that overlap with the IDRs. We demonstrated that homorepeats of disorder-promoting AAs tend to be longer in regions annotated as IDRs. However, also polyH and polyN repeats appear unexpectedly frequently as long stretches, even though amino acids H and N are of low disorder propensity. Current knowledge about the structural properties of homorepeats [25] is consistent with the idea that homo TRs play crucial roles in protein-protein interaction and signaling as well as the idea of folding in specific tertiary structures (i.e., amyloids).

Not only that length variation in polyN and polyQ stretches plays important roles in genetic diseases, but also is thought to be involved in the adaptation of organisms to their environments [76,77,78]. It was shown that polyQ and polyN rich proteins can be misfolded into amyloids which become toxic [79,80,81] and that polyQ repeats stabilize protein interactions [82]. Consistent with other studies [82,83], we found homorepeats more frequently in Eukaryotes than in other organisms. This also supports the suggestion by Schaefer et al. [82] that proteins with long polyQ stretches are enriched in more complex species with a high amount of proteins involved in protein interaction signaling and interact with proteins involved in transcriptional regulation. Finally, this agrees with our generalized view that TRs are often involved in transcriptional regulation and protein-protein interaction.

Furthermore, excellent binding properties were attributed to polyH [84], which can be finely regulated due to resulting physical and electrochemical properties such as the ability to change its charge, building charge gradients in β-sheets and has metal ion-binding. PolyH repeats accumulate in the nuclear speckle involved in transcription processes [84]. Given that TRs in general are involved in many transcriptional processes, an increased frequency of long polyH stretches can be explained through their versatility and possibility for tight regulation of transcriptional processes in the cell cycle independent of their disorder propensity which can be fine-tuned inter alia through repeat expansion.

To conclude, thanks to the new more powerful TR detection and more detailed TR annotation approaches, our study provides new insights on the universe of proteins with TRs across the kingdoms of life. While many more protein TRs are detected, our analyses in general support previous conclusions, such as Marcotte et al. [18]. However, our study shows vast variability of TR characteristics, their structures and functional roles. This heterogeneity warns against overgeneralization of common trends. Different biological processes may significantly contribute to TR origin, fixation and evolutionary mode, resulting in many exceptions from the general trend. We provide all TR and IDR annotations resulting from our study, which should enable follow-up studies focusing of specific types of TRs or proteins. This approach should help to list and start to disentangle the variety of sophisticated roles of protein TRs.

## Figures and Tables

**Figure 1 genes-11-00407-f001:**
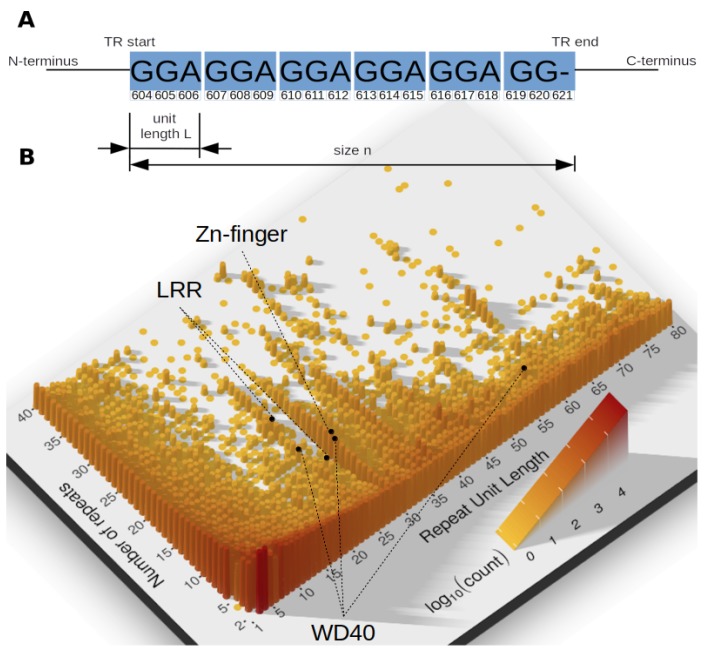
(**A**) An example tandem repeat (UniProtKB entry ID = A7TKR8) of size *n* = 6 units, each of *L* = 3 amino acids, in a head-to-head orientation. (**B**) Distribution of all annotated tandem repeats (TRs) as a function of TR unit length L≤80 and number of repeat units n≤40. Darker color and increased pillar height indicates a larger number of TRs. Most TRs has small TR units. Yet, there is a visibly high amount of domain TRs (25<L<50), and with certain unit length, e.g., L=28, mostly zinc-finger TRs.

**Figure 2 genes-11-00407-f002:**
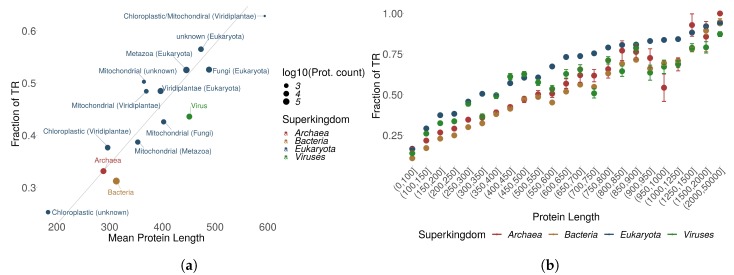
(**a**) The proportion of proteins containing at least one TR among all protein entries in UniProtKB/Swiss-Prot is shown as “proportion of TRs” for each superkingdom or kingdom, displayed as function of the mean protein length and split according to the origin of the proteins. (**b**) The proportion of proteins with TRs as a function of sequence length by superkingdom.

**Figure 3 genes-11-00407-f003:**
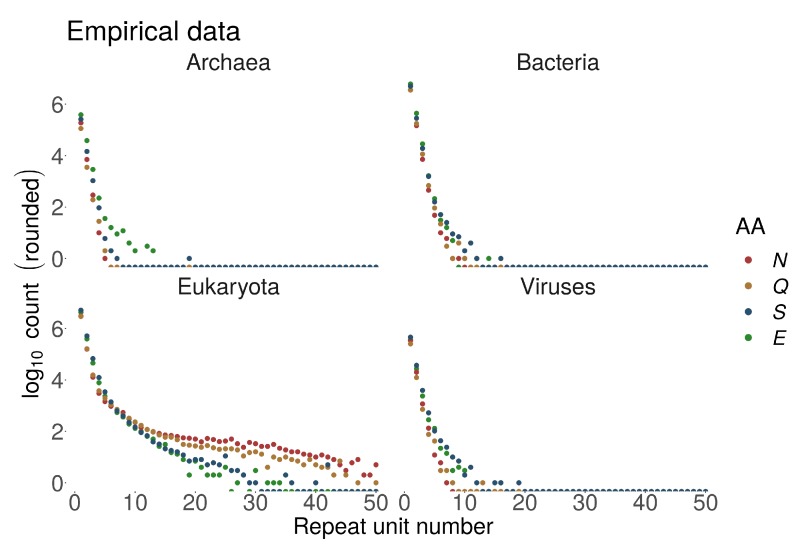
Absolute numbers of homorepeats in UniprotKB/Swiss-Prot in four superkingdoms for different number of repetitions (n≤50, equivalent to repeat length) for hydrophilic Asparagine (N), Glutamine (Q), Serine (S) and Glutamic acid (E).

**Figure 4 genes-11-00407-f004:**
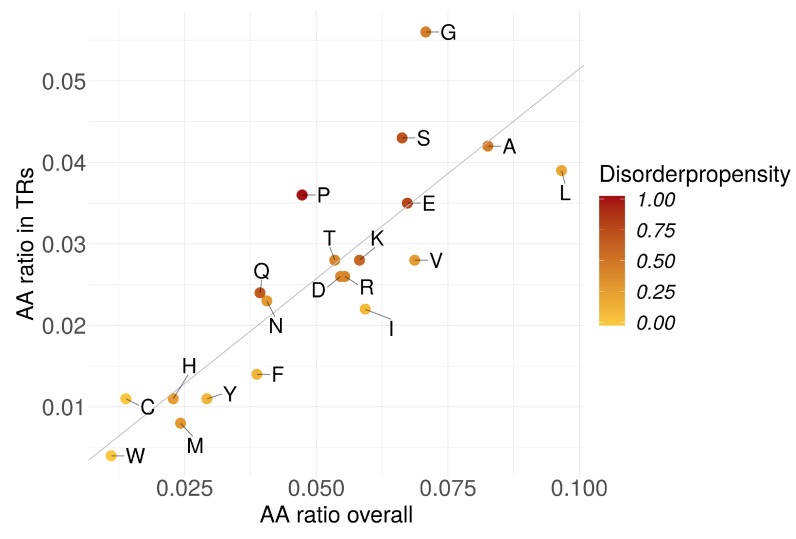
Amino acid frequencies in TRs vs. all proteins normalized by total AA numbers. Darker color indicates higher disorder propensity.

**Figure 5 genes-11-00407-f005:**
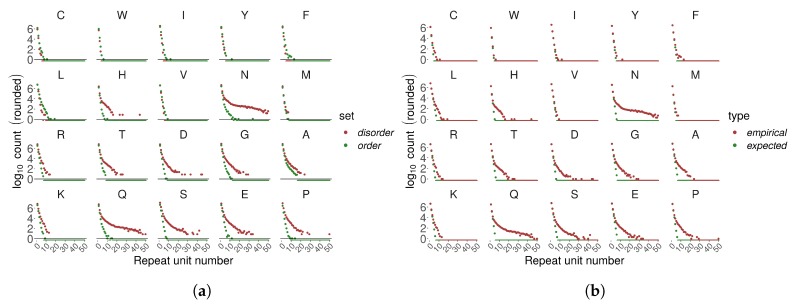
Homorepeat count in Eukaryotic proteins of UniprotKB/Swiss-Prot (n≤50). AAs are ordered by increasing disorder propensity (top-left to bottom-right). (**a**) Empirical count compared for regions annotated as disordered vs. ordered (consensus MobiDB annotations, no minimum length cut-off). (**b**) Total empirical count compared to expected.

**Figure 6 genes-11-00407-f006:**
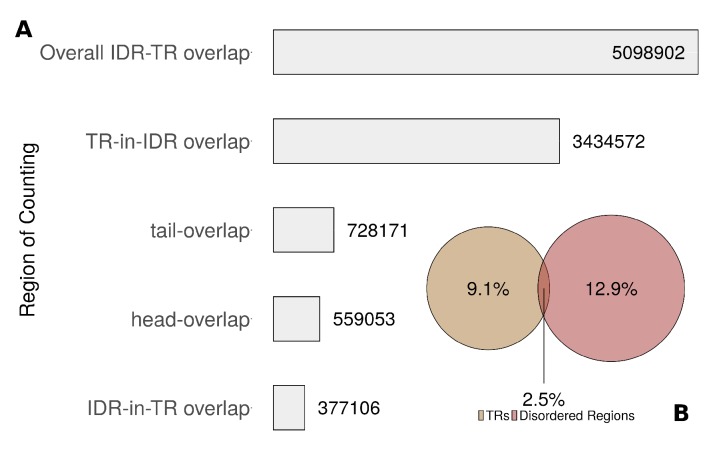
(**A**) Frequency of overlap between TRs and IDRs by type, as an absolute number of residues in all UniprotKB/Swiss-Prot. (**B**) Percentage of residues in UniprotKB/Swiss-Prot found in TRs and/or IDRs. The areas of the circles represent the total AA number in each set.

**Figure 7 genes-11-00407-f007:**
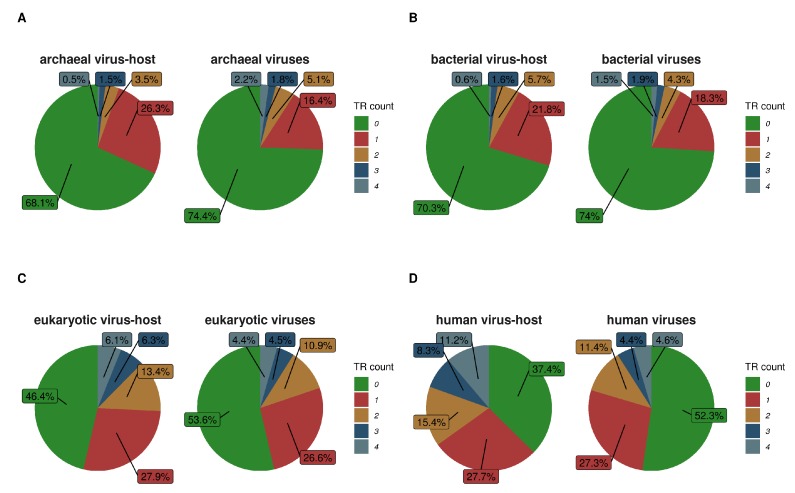
The ratio of the amount of TR per protein is shown as the number of TRs per protein divided by the total amount of proteins per group. Plots (**A**–**D**) are respectively for archaeal, bacterial, eukaryotic and human virus proteins in comparison to their associated hosts.

**Table 1 genes-11-00407-t001:** Numbers of UniProtKB/Swiss-Prot entries by superkingdoms for all proteins and for proteins that contain TR. The total amount of proteins per superkingdom is given as ‘Protein count’. ‘TR count’ refers to the number of proteins which contain at least one TR.

All Proteins	Archaea	Bacteria	Eukaryota	Viruses
Protein count	19,370	332,327	181,814	16,605
Mean protein length	288	313	436	451
TR count	6420	103,842	92,472	7237
TR fraction	0.331	0.312	0.509	0.436
homo TR	0.006	0.006	0.086	0.029
micro TR	0.117	0.109	0.245	0.191
small TR	0.217	0.208	0.328	0.300
domain TR	0.051	0.049	0.143	0.069
**Proteins Containing TRs**				
Mean protein length	355	404	572	644
Mean TR length	9.9	9.4	10.8	7.6
TR fraction				
homo TR	0.019	0.019	0.169	0.067
micro TR	0.354	0.350	0.482	0.438
small TR	0.656	0.667	0.644	0.689
domain TR	0.154	0.157	0.281	0.158

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
