# Peer review of "A New Census of Protein Tandem Repeats and Their Relationship with Intrinsic Disorder"

_genes, 2020, doi:10.3390/genes11040407_

Round 1

Reviewer 1 Report

Delucchi and colleagues present a comprehensive analysis of protein tandem repeats (TRs) across all domains of life. This is a well-performed and interesting study and the figures and tables are informative and well crafted. I have only a few minor concerns and suggestions:

1) Perhaps my biggest concern is the minimal definition of a TR. If I understood well from the text and Figure 1B, two repeated aminoacids together would be considered a TR? This would seem a bit strange to me, as I would expect this to very commonly happen just by chance. I think the authors should define clearly from the beginning what is the minimal TR and then clarify how this may affect their main claims (especially the % of proteins with PRs, etc.).

2) Overall, the authors discuss more extensively homorepeats and domain repeats than micro and small repeats. However, the latter two are the most common ones. Could they provide some insights about those? E.g. the types of motifs, if there are differences on them across domains of life, etc.

3) In the section 2.3, I think it would be very helpful that the authors exemplify the remarkable patterns (e.g. proteins with >= 4 distinct TR regions) with one or two specific cases in a figure.

4) The authors say twice in the Introduction that they have used "state of the art methods". This is vague and repetitive. Either provide info or remove.

5) Typo: line 392: bit => but.

Author Response

1) Perhaps my biggest concern is the minimal definition of a TR. If I understood well from the text and Figure 1B, two repeated aminoacids together would be considered a TR? This would seem a bit strange to me, as I would expect this to very commonly happen just by chance. I think the authors should define clearly from the beginning what is the minimal TR and then clarify how this may affect their main claims (especially the % of proteins with PRs, etc.).

OUR REPLY: We extended the Introduction (paragraph 1.1) with more details about our TR definition. The more extensive TR definition is  provided in the methods section, together with the reference to an earlier publication which explains further detail. That publication also states that for homorepeats, a separate criterion is used - only those repetitions that are not likely by chance are considered as TRs (Schaper et al 2012). So the reviewer is right, according to the full definition, 2 repetitions of one amino acid will not be considered a TR.

2) Overall, the authors discuss more extensively homorepeats and domain repeats than micro and small repeats. However, the latter two are the most common ones. Could they provide some insights about those? E.g. the types of motifs, if there are differences on them across domains of life, etc.

OUR REPLY: We did not focus on the micro and small repeats because we could not observe anything new or interesting. For example, in the two figures in the supplemental material, no special patterns can be observed .

3) In the section 2.3, I think it would be very helpful that the authors exemplify the remarkable patterns (e.g. proteins with >= 4 distinct TR regions) with one or two specific cases in a figure.

OUR REPLY: We thank the reviewer for this comment and added a short paragraph listing a few examples:

“...The most unique TRs could be annotated to structural (i.e. Titin, Fibroin, Mucin, Desmoyokin) and LRR-containing eukaryotic proteins, to pathogenesis associated bacterial proteins responsible for bacterial binding i.e. to platelets (Serine-rich adhesin for platelets) and to fibronectin (Extracellular matrix-binding protein ebh), to nuclear antigen, transcription factor and structural viral proteins and to archaeal proteins involved in mucin and DNA break repair. …”

Since our manuscript already includes many figures, we decided to not add more figures. 

4) The authors say twice in the Introduction that they have used "state of the art methods". This is vague and repetitive. Either provide info or remove.

OUR REPLY: Thank you, we removed it where it was not appropriate.

5) Typo: line 392: bit => but.

OUR REPLY: Fixed.

Reviewer 2 Report

The authors provide a census of protein tandem repeats which was certainly overdue. A previous study was published in 1999 (Reference 18); since then the number of proteins in the Swiss-Prot database has increased seven fold.

The authors provide a statistical analysis of all the proteins presently available from different points of view. Their results confirm the trends previously observed by many authors, as described in the references given in the paper, but little new insight emerges from the new data obtained.

The main contribution of the paper is a thorough analysis of the relation between tandem repeats and internal disorder in proteins, as described in sections 2.6 and 2.7.

At the end of the discussion, the authors state that they provide “all TR and IDR annotations”, but I have not found such annotations in their Github file. Also no supplementary material with such a detailed description is available. The authors should provide excel or similar files where their data could be searched by individual proteins type, protein family, repeat type, species, etc: an expanded view of the data shown in Figure 1 and in Table A1. The authors should make available to other scientists their detailed study. This will be a very useful contribution of this work to the scientific community.

The authors should also provide a clear definition of tandem repeats. How perfect are the repeats they find?

For example: AGAGAAAGAGAGAGAGTGAGAGAGAGAG could be defined either as several AG perfect repeats or a single repeat, for which a measure of internal disorder should be given,

Author Response

Reviewer 2

Comments and Suggestions for Authors

The authors provide a census of protein tandem repeats which was certainly overdue. A previous study was published in 1999 (Reference 18); since then the number of proteins in the Swiss-Prot database has increased seven fold.

The authors provide a statistical analysis of all the proteins presently available from different points of view. Their results confirm the trends previously observed by many authors, as described in the references given in the paper, but little new insight emerges from the new data obtained.

The main contribution of the paper is a thorough analysis of the relation between tandem repeats and internal disorder in proteins, as described in sections 2.6 and 2.7.

At the end of the discussion, the authors state that they provide “all TR and IDR annotations”, but I have not found such annotations in their Github file. Also no supplementary material with such a detailed description is available. The authors should provide excel or similar files where their data could be searched by individual proteins type, protein family, repeat type, species, etc: an expanded view of the data shown in Figure 1 and in Table A1. The authors should make available to other scientists their detailed study. This will be a very useful contribution of this work to the scientific community.

OUR REPLY: We agree with the reviewer and have refactored the code in the github repository such that it is easily reproducible. Our results and figures can be found mainly in the R-markdown files at /results/swissprot_*.Rmd. Additionally we added the following statement to the methods section:

“... The data analysis pipeline was implemented in R version 3.6.0 (2019-04-26) and is available on https://github.com/acg-team/swissrepeats. Therefore, the data is easily reproducible and can be used for follow up studies. …”

The authors should also provide a clear definition of tandem repeats. How perfect are the repeats they find?

For example: AGAGAAAGAGAGAGAGTGAGAGAGAGAG could be defined either as several AG perfect repeats or a single repeat, for which a measure of internal disorder should be given,

OUR REPLY:  Our approach is designed to find TRs of different degrees of perfection (i.e., TR unit divergence) since it is based on the evolutionary definition of TRs.  We extended the Introduction (paragraph 1.1) with more details about our TR definition. The more extensive TR definition is also given in the methods section together with the reference to an earlier publication which provides all details.

Round 2

Reviewer 2 Report

The changes introduced are adequate to allow final publication